# MAGNIFICO: Evaluating the In-Context Learning Ability of Large Language Models to Generalize to Novel Interpretations

**Arkil Patel**♛  **Satwik Bhattamishra**♟  **Siva Reddy**♛♕♜  **Dzmitry Bahdanau**♛♕♝

♛Mila and McGill University  ♕ServiceNow Research  ♟University of Oxford
♜Facebook CIFAR AI Chair  ♝Canada CIFAR AI Chair

{arkil.patel, siva.reddy, bahdanau}@mila.quebec  satwik.bmishra@cs.ox.ac.uk

## Abstract

Humans possess a remarkable ability to assign novel interpretations to linguistic expressions, enabling them to learn new words and understand community-specific connotations. However, Large Language Models (LLMs) have a knowledge cutoff and are costly to finetune repeatedly. Therefore, it is crucial for LLMs to learn novel interpretations in-context. In this paper, we systematically analyse the ability of LLMs to acquire novel interpretations using in-context learning. To facilitate our study, we introduce MAGNIFICO, an evaluation suite implemented within a text-to-SQL semantic parsing framework that incorporates diverse tokens and prompt settings to simulate real-world complexity. Experimental results on MAGNIFICO demonstrate that LLMs exhibit a surprisingly robust capacity for comprehending novel interpretations from natural language descriptions as well as from discussions within long conversations. Nevertheless, our findings also highlight the need for further improvements, particularly when interpreting unfamiliar words or when composing multiple novel interpretations simultaneously in the same example. Additionally, our analysis uncovers the semantic predispositions in LLMs and reveals the impact of recency bias for information presented in long contexts.

## 1 Introduction

Humans can assign new interpretations to words or phrases in a language and consequently use them compositionally in utterances. For instance, the word '*zoom*' is increasingly used to refer to a virtual calling service in the context of the COVID-19 pandemic. Similarly, as our society progresses, new words such as '*binge-watching*' and '*selfie*' keep getting coined frequently and become a part of our daily usage. Moreover, in regular conversations, people might assign custom interpretations to words or phrases (e.g., see the interpretation of '*underpaid*' in Figure 2). The question of whether

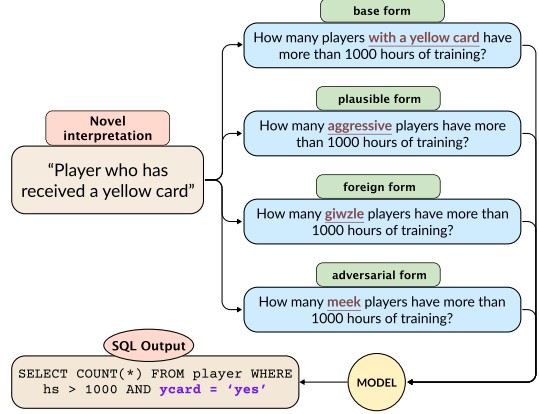

Figure 1: An example from MAGNIFICO illustrating a novel interpretation denoted by *plausible*, *foreign*, and *adversarial* forms. The corresponding *base* example is also provided.

language models are similarly capable of assigning novel interpretations to words and phrases is therefore interesting and requires investigation.

The task of learning novel interpretations has predominantly been studied from the perspective of *finetuning* a model, particularly the word embeddings, to acquire novel words during training (Lampinen and McClelland, 2017; Pham et al., 2018; Schick and Schütze, 2019). Prior studies on compositional generalization (Lake and Baroni, 2018; Kim and Linzen, 2020) also attempt to evaluate novel word learning using specially crafted *train-test splits* in which certain combinations of words are systematically held out from the test set. In recent years, however, contemporary Large Language Models (LLMs) have brought about a paradigm shift away from the classical train-test setup with their incredible capacity to learn new tasks in-context (Brown et al., 2020). With this study, we seek to understand how well can LLMs acquire novel interpretations in-context. Compared to previous setups, in-context learning (ICL) is also more practical since it is difficult to train models every time a new interpretation is encountered.

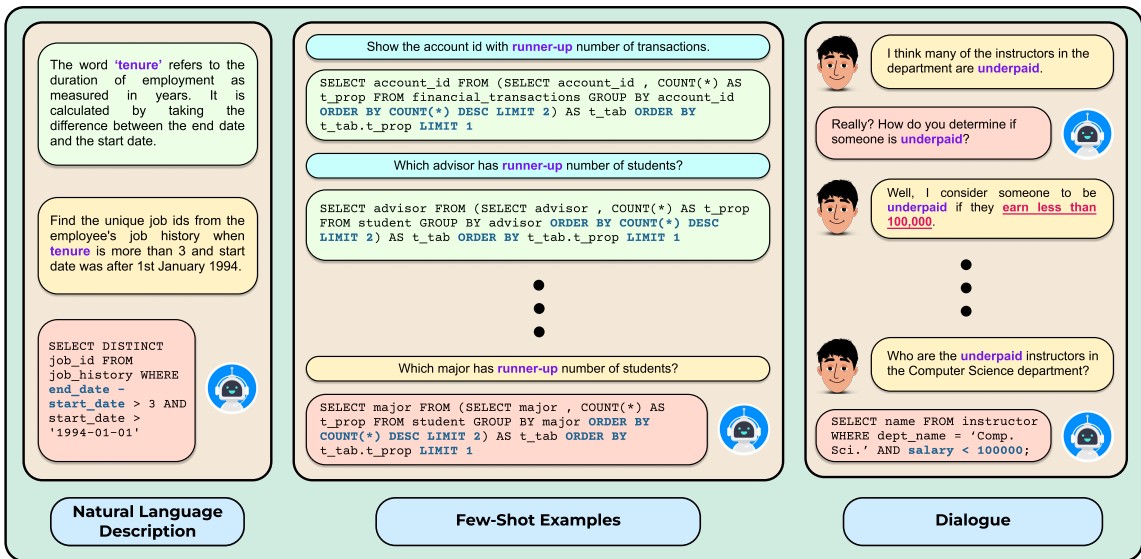

Figure 2: We consider several scenarios of how a novel interpretation can be introduced in the prompt. *Left:* A natural language description of the novel interpretation 'tenure' is provided with the question. *Centre:* Few-shot examples illustrating usage of the novel interpretation 'runner-up' are provided with the question. *Right:* Dialogue context describing the novel interpretation 'underpaid' is provided with the question.

In this work, we systematically analyse the ability of LLMs to in-context learn novel interpretations. We summarize our contributions below.

**Evaluation Suite.** To facilitate our analysis, we create an evaluation suite, **MAGNIFICO**: **M**easuring **A**daptability and **G**eneralization to **N**ovel **I**nterpretations **F**or **I**n-**Co**ntext Learning. Each example in MAGNIFICO is a text-to-SQL semantic parsing problem that requires models to understand one or more novel interpretations used in the input text to generate the correct SQL query. To simulate real-world diversity and complexity, we experiment with different ways of introducing new interpretations to the model (see Figure 2).

**Capabilities of LLMs.** We extensively experiment with 11 LLMs to understand their ability for learning novel interpretations in-context. Experiments on MAGNIFICO reveal that LLMs show a high degree of capability in learning novel interpretations even from a brief natural language description of the interpretation or from a long-form conversation. For larger LMs, learning from a description is competitive to providing explicit few-shot examples.

**Challenges for LLMs.** We find that LLMs severely fail at learning multiple novel interpretations simultaneously. Moreover, we observed that LLMs find it more challenging to learn interpretations for unfamiliar words. Our analysis also shows that LLMs have a recency bias and might find it difficult to learn interpretations presented earlier in the context.

## 2 Related Work

**Word Learning.** Previous works (Wang et al., 2017; Herbelot and Vecchi, 2016; Lampinen and McClelland, 2017; Pham et al., 2018; Schick and Schütze, 2019) have developed task- or model-specific approaches for learning the embeddings of novel words. However, these methods cannot be applied in diverse scenarios with contemporary Large Language Models (LLMs). In this work, we take a more practical approach by evaluating how well do LLMs understand novel interpretations of words and phrases *in-context* on top of a grounded NLP task, text-to-SQL semantic parsing.

There are a limited number of works that analyse the novel word learning abilities of LLMs. Haley (2020) and Thrush et al. (2020) analysed novel word learning with BERT (Devlin et al., 2019) using synthetic tests. However, it is unclear how their findings relate to autoregressive LLMs. Brown et al. (2020) qualitatively tested GPT-3's ability to use a novel word in a sentence after seeing its definition. Eisenschlos et al. (2023) analyse the in-context word learning abilities of LLMs using a synthetic co-reference resolution task. In this paper, however, we work on a more practical task and take a broader view of the problem by studying the acquisition of novel *interpretations*, which can arise even from

| CATEGORY | INTERPRETATION | EXAMPLES |
|---|---|---|
| Basic Operations | Minimum | **Input:** What are the name, latitude, and city of the station with the baseline latitude?
**Output:** SELECT name, lat, city FROM station ORDER BY lat LIMIT 1 |
| Subquery-based Operations | Most-frequent | **Input:** Display the sex and first name of students with the prevalent major.
**Output:** SELECT Sex, Fname FROM Student WHERE Major IN (SELECT Major FROM Student GROUP BY Major ORDER BY COUNT(*) DESC LIMIT 1) |
| Value-based Filtering | 4 credit courses | **Input:** List the names of all heavy courses ordered by their titles and credits.
**Output:** SELECT title FROM course WHERE credits = 4 ORDER BY title, credits |
| Column Operations | Concatenation of last and first name | **Input:** How many students are there with 'gE' in alias?
**Output:** SELECT COUNT(*) FROM student WHERE lname \|\| fname LIKE '%gE%' |

Table 1: Examples of novel interpretations in MAGNIFICO. Illustrated examples use a *plausible* English form.

existing words and phrases in the vocabulary. We also study compositional generalization of multiple novel interpretations simultaneously.

**Compositional Generalization.** Recent works (Lake and Baroni, 2018; Kim and Linzen, 2020; Keysers et al., 2020) proposed benchmarks with a systematic difference between the train and test sets: novel combinations of certain words are held out from the train set. However, such evaluation setups are susceptible to fairness issues (Sikarwar et al., 2022) arising from the dependence on a train set. Moreover, model-independent factors in the train set can influence generalization performance (Patel et al., 2022). Our evaluation framework is set within the paradigm of in-context learning (ICL), which does not require the creation of an explicit train set. Note that even in ICL settings, LLMs have saturated existing compositional generalization benchmarks (Drozdov et al., 2023). More recently, An et al. (2023) proposed a new benchmark, CoFE, for in-context compositional generalization. However, their focus was on understanding the factors affecting better selection of in-context examples for compositional generalization. Moreover, the examples in CoFE are based on another synthetic benchmark, while we focus on more realistic settings using a grounded text-to-SQL task.

**Knowledge-intensive text-to-SQL.** Works on knowledge-intensive text-to-SQL (Li et al., 2023a; Lee et al., 2021; Zhao et al., 2022; Dou et al., 2022) have some similarities with our work in that they assign schema-specific external knowledge to words or phrases in the input query. However, our interpretations are much more dynamic and do not have pre-defined formal definitions. Moreover, the focus of these works is on domain generalization for text-to-SQL semantic parsing. We only use text-to-SQL as a testbed since it allows us to more formally ground meanings of interpretations and has real-world applicability.

## 3 MAGNIFICO

We choose the text-to-SQL task to test LLMs' ability to handle novel interpretations because of its relevance to real-world applications. Moreover, contemporary LLMs already achieve good zero/few-shot in-context performance on the task. We create MAGNIFICO by modifying and re-tasking examples from an existing text-to-SQL benchmark, Spider (Yu et al., 2018). Below, we describe the procedure in detail.

### 3.1 Novel Interpretations

We define a set of 24 interpretations that are either already being used or can be introduced in the examples of Spider:

- ★ Basic operations: Standard column operations frequently used in SQL queries.

- ★ Subquery-based operations: Complex operations using nested subqueries.

- ★ Value-based filtering: Particular subset of values for specific columns.

- ★ Column operations: Operations performed over specific columns.

Table 1 provides examples of some of the interpretations that we defined. We will refer to the word or phrase used to denote the novel interpretation on the source side of the example as its *form*. We defined 18 interpretations denoted by a single word form and 6 interpretations denoted by a phrase form (see Figure 3 for an illustration). The full list of all interpretations can be found in Tables 4 and 5 in the Appendix. For the 18 interpretations denoted by a single word, we experiment with three types of forms that vary in their pre-existing semantic meaning: (1) *plausible* forms are words that can

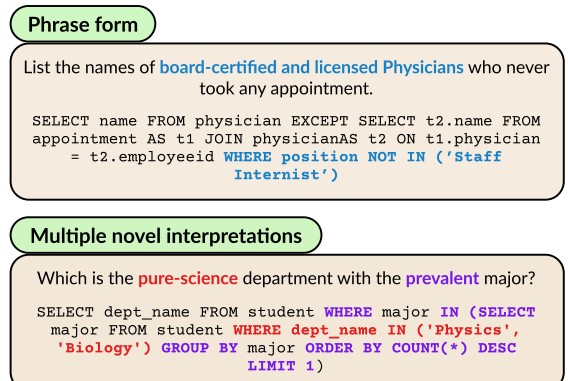

**Phrase form**

List the names of board-certified and licensed Physicians who never took any appointment.

```
SELECT name FROM physician EXCEPT SELECT t2.name FROM
appointment AS t1 JOIN physicianAS t2 ON t1.physician
= t2.employeeid WHERE position NOT IN ('Staff
Internist')
```

**Multiple novel interpretations**

Which is the pure-science department with the prevalent major?

```
SELECT dept_name FROM student WHERE major IN (SELECT
major FROM student WHERE dept_name IN ('Physics',
'Biology') GROUP BY major ORDER BY COUNT(*) DESC
LIMIT 1)
```

Figure 3: Illustrations of examples with *phrase* form and multiple novel interpretations in MAGNIFICO.

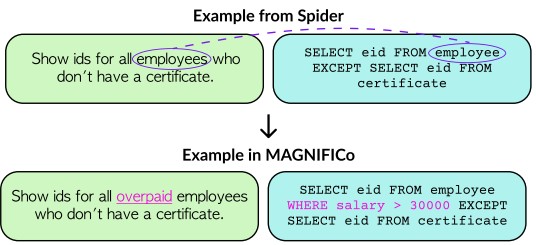

Figure 4: Illustration of the 'Regex-based pattern matching' procedure used for creating examples in MAGNIFICO.

reasonably be expected to represent the novel interpretation in realistic scenarios, (2) *foreign* forms are novel words without any pre-defined meaning that are generated using a random permutation of English characters, and (3) *adversarial* forms are words with an existing meaning that is contrary to the intent expressed by the novel interpretation. Figure 1 illustrates the three types of forms in an example from MAGNIFICO.

### 3.2 Generating Data

We create MAGNIFICO examples by modifying examples in the Spider dataset such that understanding a novel interpretation[1] used in the input is necessary to successfully generate the corresponding SQL query. We will refer to the original examples from Spider as *seed* examples. For each interpretation, we generate data using one or more of the following methods:

**(1) Regex-based pattern matching.** Some interpretations such as *'the minimum value'* (see Table 1) already have examples existing in Spider. For such interpretations, we find the relevant seed examples using regex-based pattern matching, either on the source side by conditioning on the presence of certain keywords such as *minimum* or *lowest* or on the target side by conditioning on operations such as `min()`. We then modify the seed examples to include the form of the interpretation in the input and inculcate the corresponding logic in the target SQL query using specific rules, if required. An illustration of this process is shown in Figure 4.

**(2) LLM-assisted constrained paraphrasing.** For many interpretations, it is not possible to manually devise rules for modifying the natural language

queries of the seed example to include the corresponding form in a suitable manner. In such cases, we prompt GPT-4 (OpenAI, 2023) with the query of the seed example and instruct it to paraphrase the query so as to include the form of the novel interpretation. We then manually examine the model-generated paraphrase for grammatical correctness and consistency of intended meaning. Similar to the previous method, we modify the target SQL query using hand-crafted rules, if required.

**(3) Synchronous Context-Free Grammar.** For many interpretations, there are either none or very few examples already existing in Spider. It is also difficult to find such examples automatically based on regex patterns. In such cases, if we have obtained a limited number of examples from Spider, we extract an SCFG representing those examples by abstracting away specific column and table names and data values similar to the method used by Yu et al. (2021). If there are not enough examples in Spider, we define an SCFG ourselves that represents the interpretation. We then automatically generate examples from the SCFG by filling it with randomly sampled column and table names and values from the set of databases in Spider. We only keep the examples for which the generated SQL query correctly executes and does not give a `NULL` output. An illustration of this process is provided in Figure 5.

**Multiple novel interpretations in same example.** From the data created using the procedures above, we select pairs of examples that have different novel interpretations but use the same database schema. We devise a set of rules using which, given such a pair, we can automatically create a new example that requires understanding both novel interpretations (one from each of the examples in the pair) simultaneously. Figure 3 illustrates such an example. We created a

---

[1]We experiment with different types of prompt contexts for explaining the novel interpretation, detailed in §4.

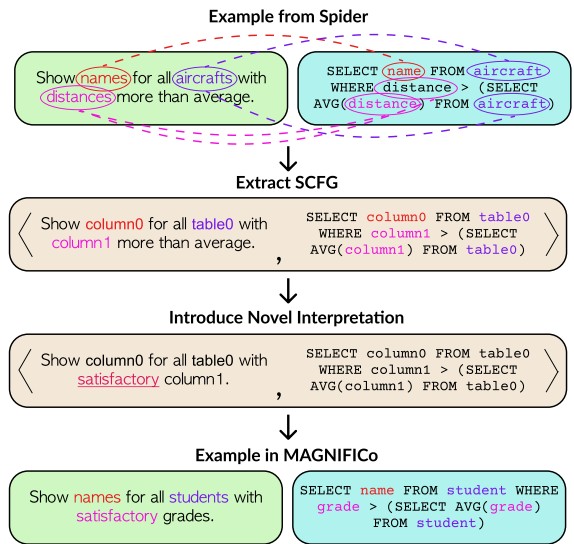

Figure 5: Illustration of the 'Synchronous Context-Free Grammar' procedure used for creating examples in MAGNIFICO.

| Type | Unique Templates | Types of Forms | Num of Examples |
|---|---|---|---|
| Single-word | 1150 | base, adversarial, plausible, foreign | 4600 |
| Phrase | 279 | base, plausible | 558 |
| Multiple novel interpretations | 94 | base, adversarial, plausible, foreign | 376 |
| Total | 1523 | | 5534 |

Table 2: Dataset statistics for MAGNIFICO.

total of 376 such examples spanning 24 unique combinations of interpretations. We manually reviewed each example to ensure correctness.

**Generating Conversations.** We generate long-form dialogues for a subset of examples in MAGNIFICO. For each database schema used in these examples, we prompt GPT-4[2] to generate a long conversation between two users of that database. We instruct GPT-4 to introduce the corresponding novel interpretation and its description in a manner that makes it blend naturally into the flow of the conversation at the beginning. We generated a total of 125 unique dialogues, each at least 2000 tokens long. We manually reviewed all generated dialogues to ensure correctness.

**Base Examples.** We are only interested in measuring the ability of models to generalize to novel interpretations and not how well they perform on the text-to-SQL semantic parsing task. Hence, for every example in MAGNIFICO with a novel interpretation, we also maintain an example that does not include any novel interpretation form and instead directly states the interpretation as part of the input query. These examples serve as a comparative reference point for our evaluation. We refer to these examples as *base* examples and measure the performance of all models across all

---

[2]Prompt provided in Figure 19 in the Appendix.

prompt types on them. An illustration of a *base* example is shown in Figure 1.

**Summary.** Overall, we created 1523 unique examples across 24 interpretations. The forms of interpretations in these examples can be automatically replaced to generate more data at scale. Dataset statistics for MAGNIFICO are provided in Table 2. Additional details on creating MAGNIFICO are provided in Appendix B. Note that each example in MAGNIFICO was manually reviewed by at least one of the authors to ensure correctness.

## 4 Experimental Setup

In this section, we will discuss the setup for our experiments on MAGNIFICO.[3]

**Models.** We experiment with OpenAI GPT-3.5-Turbo (v0301) (Brown et al., 2020; Ouyang et al., 2022), StarCoder (Li et al., 2023b), LLaMA-7B,13B,30B (Touvron et al., 2023a), Alpaca-7B (Taori et al., 2023), MPT-7B[4], MPT-7B-Instruct, RedPajama-7B[5], RedPajama-7B-Instruct, and RWKV-14B (Bo, 2021). We additionally experimented with GPT-4 (OpenAI, 2023) and LLaMA-2 (Touvron et al., 2023b), results for which are provided in Appendix C.4. For all models, we decode greedily for a maximum of 128 tokens. To take stock of the basic text-to-SQL semantic parsing capabilities of these models, we show their execution accuracies over the base examples in MAGNIFICO averaged over all interpretations in Figure 6. Some of the results for RWKV-14B and the RedPajama-7B models can be found in Appendix C.

**Prompt.** All our experiments are in the in-context learning experimental setup. Our prompt structure

---

[3]We make our code and data available at https://github.com/McGill-NLP/MAGNIFICO.

[4]https://www.mosaicml.com/blog/mpt-7b

[5]https://www.together.xyz/blog/redpajama-models-v1

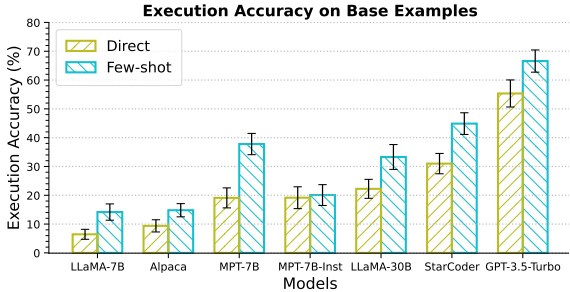

Figure 6: Average execution accuracy (↑) of all models on *base* examples in MAGNIFICO across various prompt settings.

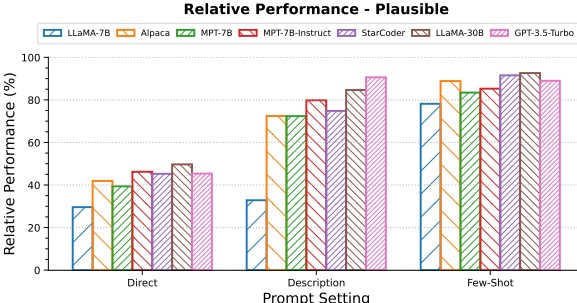

Figure 7: Relative performance (↑) of all models on MAGNIFICO across various prompt settings when the TOKEN is a *plausible* English word.

largely follows the *Create Table + Select 3* prompt format from Rajkumar et al. (2022) which resulted in the best performance on Spider with OpenAI Codex (Chen et al., 2021). This format provides the CREATE TABLE commands for each table in the schema and displays the values for the top three rows for each table. We experiment with three different prompt settings: (1) **'Direct'** is exactly the zero-shot *Create Table + Select 3* setting which includes no information about how to interpret the form of the novel interpretation, (2) **'Description'** additionally includes a brief, one-line natural language description of the novel interpretation(s), and (3) **'Few-shot'** includes 5 input-output examples[6] of the novel interpretation instead of the description. We hold out 5 examples for each interpretation from the test sets to maintain consistency in testing across various experimental settings. For experiments with a dialogue in the context, the dialogue is prepended to the *Create Table + Select 3* prompt. Examples for each type of prompt are provided in Appendix E.

**Metric.** We use a metric called *Relative Performance* to measure the generalization ability of models towards acquiring novel interpretations. Our metric provides a measure that is relative to the performance of the model on the corresponding *base* examples:

$$\text{Relative Performance} = min\left(\frac{\text{EX}^{\text{NI}}}{\text{EX}^{\text{base}}}, 1\right) \times 100$$

where $\text{EX}^{\text{NI}}$ is the execution accuracy[7] on the examples with novel interpretations from MAGNIFICO and $\text{EX}^{\text{base}}$ is the execution accuracy on

the corresponding base examples.[8] Hence, the higher the *Relative Performance*, the lower the model's drop in performance on examples with novel interpretation(s) (relative to base examples), and consequently, the higher its ability to learn novel interpretations.

## 5 Results and Discussion

### 5.1 Impact of Description and Few-shot Examples

*Question: How well can LLMs learn novel interpretations when the interpretation is simply described in an English sentence? And how does it compare against the case when we provide few-shot examples of usage of that interpretation?*

We compare providing a natural language description of the novel interpretation (**'Description'** prompt type) against providing examples of usage of the novel interpretation (**'Few-shot'** prompt type). Figure 7 provides the results when the form used to represent the novel interpretation is a *plausible* English word. The results for *foreign* and *adversarial* forms can be found in Figures 16 and 17 in the Appendix.

Most LLMs exhibit a surprisingly high capability to understand and generalize to novel interpretations from simply the natural language descriptions. This capability seems to increase with model size as GPT-3.5-Turbo and LLaMA-30B outperform all other models while the smallest model, LLaMA-7B, struggles to generalize from just the description. It is also interesting to see the benefit of instruction finetuning (Wang et al., 2023) in learning novel interpretations in-context just from natural language descriptions: the instruction-finetuned models outperform their corresponding base models, often by

---

[6]For multiple novel interpretations in the same example, we include 3 support examples for each novel interpretation.

[7]Measure of equivalence between output obtained from executing the generated SQL query and the ground truth output.

---

[8]We only consider interpretations for which the execution accuracy on base examples is at least 5%.

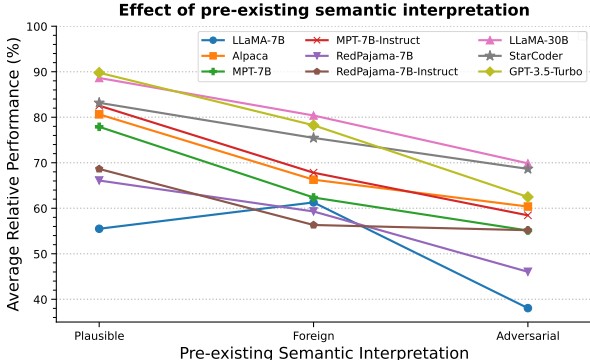

Figure 8: Relative performance (↑) of all models across forms of different types.

| **Prompt** | StarCoder | | GPT-3.5-Turbo | |
|---|---|---|---|---|
| | Plausible | Foreign | Plausible | Foreign |
| Description | 79.40 | 80.74 | 91.46 | 85.95 |
| Dialogue | 68.91 | 80.55 | 84.87 | 87.63 |

Table 3: Relative performance (↑) of StarCoder and GPT-3.5-Turbo on examples in MAGNIFICo when the description of the novel interpretation is provided in a long form dialogue.

large margins. All models generalize well when a few examples of usage of the novel interpretation are provided.

## 5.2 Impact of Pre-existing Semantic Interpretation

*Question: How much does the existing semantic interpretation of the form denoting the novel interpretation influence the generalization capability of LLMs?*

As mentioned in §3.1, we experiment with three types of form. We plot the relative performance averaged over the **'Description'** and **'Few-shot'** prompt types[9] in Figure 8.

We see a trend of decrease in generalization ability when the pre-existing semantic interpretation of the form steers farther from the intended meaning of the novel interpretation. This shows that LLMs have strong semantic priors that may require targeted approaches to be overcome. Moreover, the fact that LLMs can easily understand novel interpretations when presented in a familiar form (as opposed to completely foreign words) is an interesting finding for potential applications requiring acquisition of novel interpretations in the wild (e.g., conversational agents).

## 5.3 Acquiring Novel Interpretations From Long Form Dialogue

We envision a real-life scenario requiring compositional generalization: acquiring novel interpretations introduced in a long form conversation. This may arise in situations such as having a conversation with an AI personal assistant or when we want to condition the outputs of an AI system based

on a dialogue history between multiple users. An example is provided in Figure 2.

*Question: How well can LLMs learn a novel interpretation from its description mentioned briefly within a long-form dialogue?*

We select 8 interpretations, covering a total of 583 examples from MAGNIFICo, encompassing all four categories. We generate long conversation contexts for each of these examples as described in §3.2. An example of the prompt structure is provided in Figure 22. We experiment with StarCoder and GPT-3.5-Turbo since they are capable of processing more than 2000 tokens of text in-context. The results are provided in Table 3. For ease of comparison, we also state the results with the 'Description' prompt-type for the 8 interpretations considered.

For both models, using a *foreign* form to represent the novel interpretation does not result in much performance difference when the description of the novel interpretation is blended inside a long form dialogue instead of directly stating it. However, when the form is a *plausible* english word, we see a clear decrease in generalization ability for both models. The decrease is much more significant for StarCoder compared to GPT-3.5-Turbo. This indicates that LLMs may find it difficult to associate a case-specific interpretation with tokens that they are already familiar with when used in long conversations. It is possible that the models do not pay much attention to that aspect of the conversation as it might seem more 'normal' compared to the case where a *foreign* form is used.

## 5.4 Impact of Position of Description in Context Window

*Question: How sensitive are LLMs to the location in the context window that the novel interpretation is described in?*

We experiment with placing the description at the beginning, middle, or the end of the prompt

---

[9]We average over the prompt types to improve readability of the figure. The complete figure can be seen in Figure 18 in the Appendix.

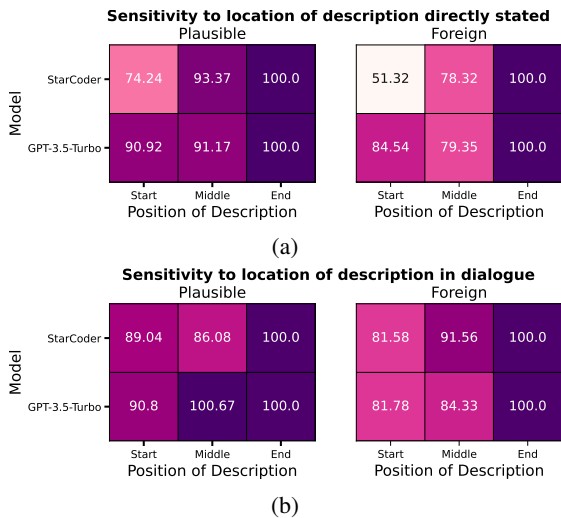

(a)

(b)

Figure 9: Relative performance (↑) of `StarCoder` and `GPT-3.5-Turbo` on MAGNIFICo across different locations of the description of the novel interpretation in the prompt when the description is directly stated (*top*) and when the description is mentioned in a dialogue (*bottom*). The numbers are relative to the performance for the *end* position.

when using the **'Description'** prompt type. We also experiment with the **'Dialogue'** setting by placing the turns of conversation describing the novel interpretation at the beginning, middle, or the end of the dialogue. The results for both experiments are provided in Figure 9. Note that we measure performance relative to the performance when the novel interpretation is described in the end so as to better characterize recency bias.

We observe a clear trend of recency bias in both LLMs, where the generalization increases when the interpretation is described nearer to the end of the context window. `StarCoder` suffers much more variation in performance compared to `GPT-3.5-Turbo`. The difference in performance between *start* and *end* positions for `GPT-3.5-Turbo`, while comparatively small, is still significant enough to indicate a stronger preference for information presented later in the context.

## 5.5 Composing Multiple Novel Interpretations

*Question: Are LLMs able to simultaneously learn multiple novel interpretations used compositionally in the same example?*

We evaluate models on a total of 376 examples that require simultaneously understanding two novel interpretations (see Figure 3 for an example). The results for all models across all three types of form of interpretations using the **'Description'** and

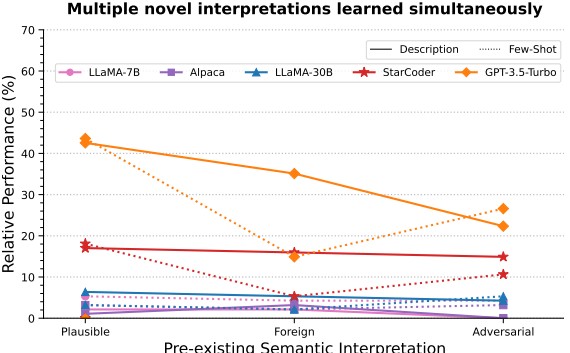

Figure 10: Relative performance (↑) of all models across all settings when there are multiple novel interpretations in the same example.

**'Few-shot'** prompt types are provided in Figure 10.

We notice that all models struggle at learning multiple novel interpretations in the same example compared to learning just one novel interpretation. `GPT-3.5-Turbo` is the best performing model, significantly outperforming `StarCoder` while the rest of the models show nearly trivial performance. The difference in performance between 'description' and 'few-shot' prompt types for *foreign* form suggests that models have a comparatively harder time composing interpretations when they are presented individually in separate examples in the prompt.

## 5.6 Learning Novel Interpretations of Phrases

*Question: Are LLMs able to learn novel interpretations when they are denoted by more than a single word?*

We defined 6 interpretations denoted by phrases of plausible English words in MAGNIFICo, amounting to a total of 279 examples (see Figure 3 for an example). The results of evaluation over these examples are provided in Figure 11.

We notice that LLaMA, `StarCoder`, and `GPT-3.5-Turbo` models show a surprisingly high ability to learn the novel interpretation from just the description. It is even more surprising to see both `MPT-7B` models struggle since they comparatively excelled for single-word form interpretations (see Figure 7). This shows that the task of learning novel interpretations denoted by multi-word phrases is not simply an extension of learning single-word form interpretations, but a separate task that presents its own set of challenges. Lastly, it is interesting to see that contrary to expectations, `StarCoder` outperforms `GPT-3.5-Turbo` in both prompt settings.

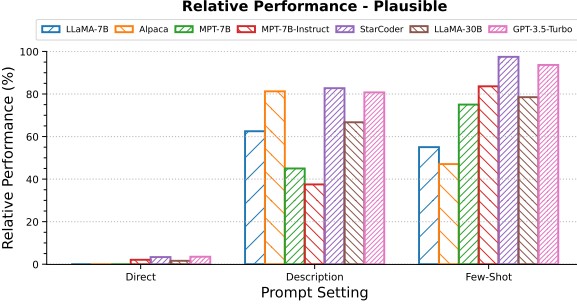

Figure 11: Relative performance (↑) of all models on MAGNIFICO across various prompt settings when the novel interpretation is a phrase composed of multiple English words.

## 6 Final Remarks

We studied the ability of LLMs to interpret new words and phrases in-context using their description or a few demonstrations. We also extended this study to a realistic scenario: understanding user-defined interpretations from long form dialogue.

Our results indicate that current LLMs can, to an extent, acquire novel interpretations from diverse forms of context. However, interpreting unfamiliar words or multiple novel words simultaneously still poses a significant challenge for existing LLMs. These tasks can serve as a measure to evaluate the compositional generalization abilities of future LLMs in practical settings.

It is interesting to note that instruction fine-tuning leads to significant improvements in learning from descriptions across three different LLM families. Considering that instruction fine-tuning doesn't involve acquiring novel semantics, it could be useful to understand why it has this impact.

In the past few years, several works (Lake and Baroni, 2018; Kim and Linzen, 2020) showed that sequence models were limited in their ability to generalize to novel words on semantic parsing tasks based on a few examples in the training set. Many specialised methods and approaches (Liu et al., 2021; Chen et al., 2020) were designed to address this problem. It is therefore fascinating to see that contemporary *general* LLMs are able to generalize to novel words from not only processing a few examples in-context, but also from natural language descriptions and conversations. While a large part of the compositional generalization challenge still remains unsolved, we feel it is important to highlight this paradigm shift. We hope our work paves the way for further studies of practical setups that require LLMs to generalize compositionally.

## Acknowledgments

We would like to thank Navin Goyal for initial discussions related to the idea behind this work. We also thank the anonymous reviewers, and our colleagues at Mila and McGill University for helpful discussions and for providing valuable feedback. Arkil is also supported by the Canada Graduate Scholarship – Master's (CGS-M) funded by the Natural Sciences and Engineering Research Council of Canada (NSERC).

## Limitations

We created our evaluation suite MAGNIFICO over a single task, text-to-SQL semantic parsing. While semantic parsing is a fundamental task in language processing with general applicability, it would be useful to verify the findings across other tasks and domains. In the future, we aim to incorporate more tasks from diverse domains such as classification to better support our claims.

The execution accuracies over *base* examples in MAGNIFICO are low for smaller models. This results in a higher variance in the results of small models. While we enforce a threshold of minimum 5% accuracy on the base examples for each interpretation to be included in the results, in the future, we shall also include experiments over a task that is more easily solvable by smaller models.

The number of data points for some of our experimental settings (such as multiple novel interpretations) is not large. However, note that our study was exploratory in nature and our main focus was on *analysing* the in-context learning abilities of LLMs for acquiring novel interpretations rather than proposing a general benchmark for evaluating LLMs. Our findings revealed that LLMs face more difficulty when there are multiple novel interpretations in the same example. This motivates us to look more closely at this particular setting in the future and potentially create a challenging benchmark.

## Ethics Statement

We have extensively discussed the limitations of our work in the previous section. We use an existing dataset, Spider (Yu et al., 2018), which is publicly available and commonly used in NLP research. We generate additional data by modifying the examples in Spider in a rule-based manner. Since we focus on a text-to-SQL semantic parsing task, there

are minimal risks and biases associated with our data and we believe that it does not require ethical consideration. We also employed Large Language Models to automatically generate data, and each example of the generated data went through manual verification, ensuring that it does not pose any significant risk. We have discussed the experimental details in Appendix A. The research presented in this paper focuses on analysing the in-context learning abilities of LLMs targeted towards interpreting novel interpretations and we believe that our work does not raise any ethical concerns.

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

## A   Implementation Details

Experiments using `GPT-3.5-Turbo` and `GPT-4` were performed using the OpenAI API[10]. All other experiments were done on a single NVIDIA A100 GPU with 80 GB memory. Our code is implemented in PyTorch (Paszke et al., 2019) and makes use of the HuggingFace Transformers library (Wolf et al., 2020).

## B   Additional Information on MAGNIFICo

We provide examples for each of the 18 single-word form and 6 phrase form interpretations in MAGNIFICo in Table 4 and Table 5 respectively.

### B.1   Populating Tables with Edge Cases

The metric for evaluating the generated SQL queries in text2SQL benchmarks is execution accuracy, which compares the output of the execution of the generated query with the ground truth. Since we are introducing new interpretations in existing databases, it is possible that the output of the corresponding SQL query is trivial, like printing all values in the Table. Apart from this, it is possible that an incorrect SQL query leads to the ground-truth output because there are no edge case values present in the Table. To handle such cases, we automatically populate the tables by inserting new values that act as edge cases (Zhong et al., 2020).

## C   Additional Experimental Results

### C.1   Performance on Base Examples

The performance of models on *base* examples in MAGNIFICo can be seen in Figure 12. We found the base text-to-SQL performance of `RWKV-14B` to be extremely low and hence do not experiment with it in other settings.

### C.2   Impact of Description and Few-Shot Examples

Figure 7, Figure 16 and Figure 17 illustrate the impact of providing description and few-shot examples of the novel interpretation when the novel interpretation is represented by a *plausible*, *foreign* or an *adversarial* form respectively for all models.

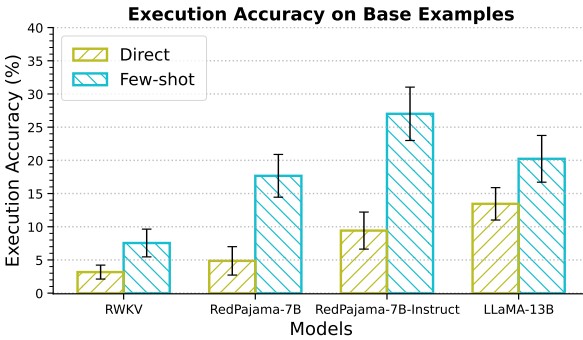

Figure 12: Average execution accuracy (↑) of models on *base* examples in MAGNIFICo across various prompt settings.

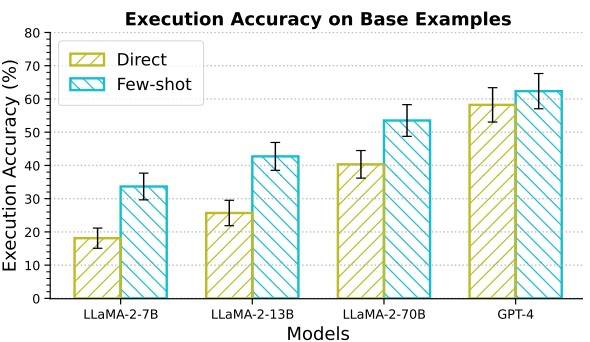

Figure 13: Average execution accuracy (↑) of `LLaMA-2` and `GPT-4` models on *base* examples in MAGNIFICo across various prompt settings.

### C.3   Impact of Pre-existing Semantic Interpretation

Figure 18 provides the results for all models across all experimental settings.

### C.4   Results for LLaMA-2 and GPT-4

The performance of `LLaMA-2` and `GPT-4` models on *base* examples in MAGNIFICo can be seen in Figure Figure 13. Their performance across all experimental settings can be seen in Figure 14.

## D   Additional Related Works

**Word Acquisition**

Lazaridou et al. (2021) analyse the temporal generalization capabilities of LLMs and showed that the perplexity increases when modelling text containing new words. There is also some related work in the domain of grounded language learning. Chevalier-Boisvert et al. (2019) focus on learning a synthetic language which is a subset of English. However, they do not carry out any systematic evaluation focused on word learning. Hill et al. (2021) propose an approach for *fast-mapping*, i.e.,

---

[10]https://platform.openai.com/

the ability to bind a novel non-sense word to an object in their RL framework. However, their framework and approach are specifically designed to cater to word learning, while we wish to evaluate the word learning abilities of general NLP models across various NLP tasks. Tsimpoukelli et al. (2021) focus on using frozen pretrained models for learning words that only act as names of objects in images. We wish to study word learning at a broader level, by considering more complex types of words and interpretations.

**Compositional Generalization**

Many works in the past (Fodor and Pylyshyn, 1988; Hadley, 1994; Fodor and Lepore, 2002; Marcus, 2003; Calvo and Symons, 2014) have argued that artificial neural networks are incapable of exhibiting systematic compositionality. However, recent successes of neural models (Bahdanau et al., 2015; Vaswani et al., 2017; Devlin et al., 2019) across various NLP tasks have revived this debate with a focus on investigating the presence and extent of compositional biases in models.

Lake and Baroni (2018) investigated the compositional generalization abilities of contemporary neural sequence models such as RNNs and LSTMs based on their performance on a synthetic benchmark called 'SCAN'. Their conclusions were consistent with past work in that they found neural sequence models generalize poorly when tested on systematically held-out novel combinations of words and phrases. Follow-up work by Kim and Linzen (2020) reached similar conclusions using their semantic parsing benchmark, 'COGS'.

While novel word learning has not been explicitly studied in previous compositional generalization literature, some of the experiments carried out by Lake and Baroni (2018) and Kim and Linzen (2020) do implicitly assess the abilities of models to one-shot acquire a novel word. However, the words used in these experiments are of a *primitive* nature and have a context-independent direct mapping in the output space (for e.g., in SCAN, models simply need to learn to map the input word 'jump' to its corresponding output token 'JUMP'). In our work, we broaden the scope to also understand how well models acquire more *functional* words, i.e., words that act over other words in a context-dependent manner to generate the output (for e.g., consider the interpretation 'most-frequent' represented by the form *prevalant* in Table 1. The output looks

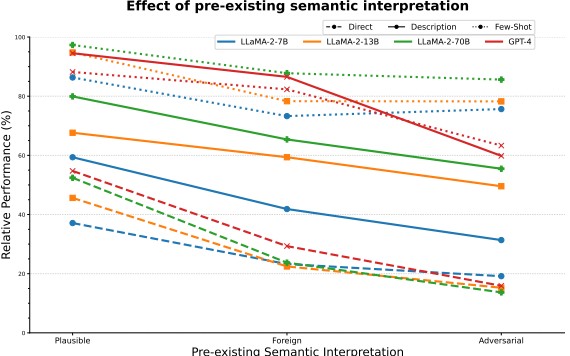

Figure 14: Relative performance (↑) of LLaMA-2 and GPT-4 models across forms of different types for all prompt settings.

very different for inputs like, 'Find the prevalant age of students' or, 'What is the number of students that do not have the prevalant last name?').

There have been many compositional generalization benchmarks proposed in recent years (Keysers et al., 2020; Yanaka et al., 2021), almost all of them illustrating deficiencies of neural models at generalizing compositionally. Many approaches have also been proposed to solve compositional generalization benchmarks (Li et al., 2019; Lake, 2019; Gordon et al., 2020; Chen et al., 2020; Andreas, 2020; Liu et al., 2020; Guo et al., 2020; Akyurek and Andreas, 2021; Conklin et al., 2021; Liu et al., 2021). However, most of these approaches are task-specific and cannot be generally applied for language processing. Moreover, LLMs achieve a very high level of performance on compositional generalization benchmarks based on just a few examples in-context (Drozdov et al., 2023). In this work, we seek to analyse the compositional generalization capabilities of LLMs more realistically, by grounding our evaluation to possible use case scenarios, for e.g. generating SQL queries for user inputs that require understanding novel interpretations from a long conversation context.

# E    Example Prompts

Figure 19 shows an example of a prompt used to generate a long form dialogue using GPT-4. Figures 20, 21, and 22 show examples for the **'Description'**, **'Few-shot'**, and **'Dialogue'** prompt types respectively.

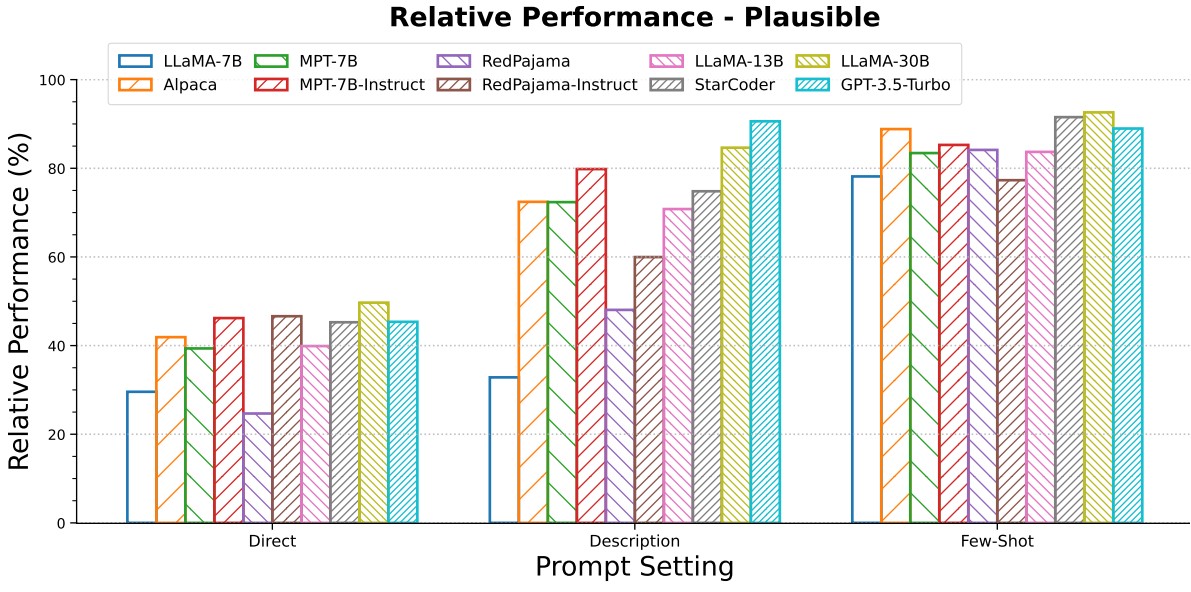

Figure 15: Relative performance (↑) of all models on MAGNIFICO across various prompt settings when the TOKEN is a *plausible* word.

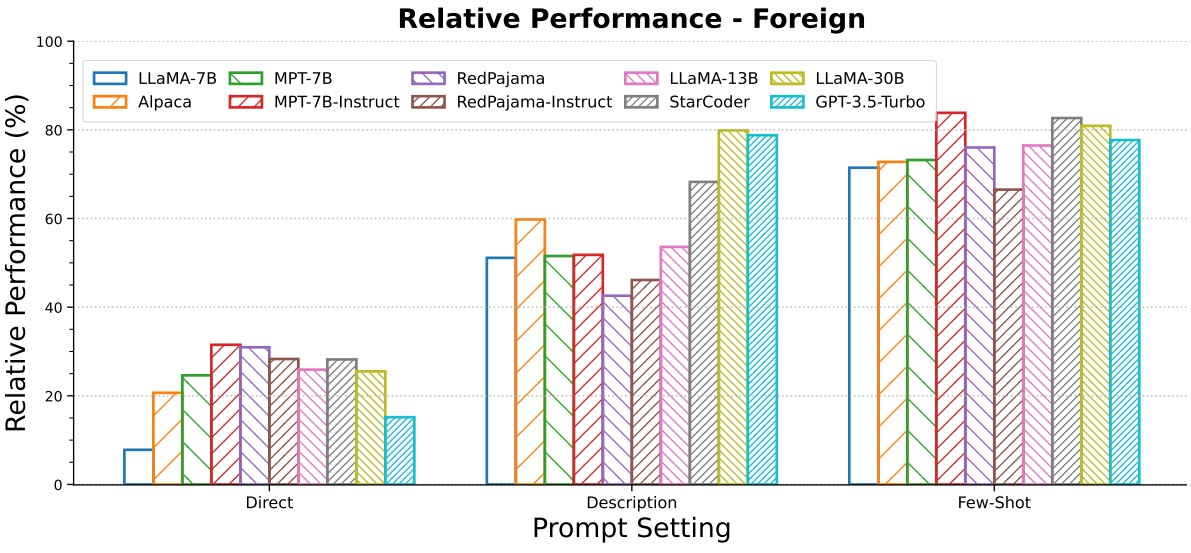

Figure 16: Relative performance (↑) of all models on MAGNIFICO across various prompt settings when the TOKEN is a *foreign* word.

| CATEGORY | INTERPRETATION | EXAMPLES |
|---|---|---|
| Basic Operations | Minimum | **Input:** What are the name, latitude, and city of the station with the baseline latitude?
**Output:** `SELECT name, lat, city FROM station ORDER BY lat LIMIT 1` |
| | Maximum | **Input:** Which film has the coveted rental rate? And what is the rate?
**Output:** `SELECT title, rental_rate FROM film ORDER BY rental_rate DESC LIMIT 1` |
| | Average | **Input:** What is the representative price for flights from LA to honolulu?
**Output:** `SELECT AVG(price) FROM flight WHERE origin = 'Los Angeles' AND destination = 'honolulu'` |
| | Sum | **Input:** What is the accumulated employee number for each office of professors?
**Output:** `SELECT prof_office, SUM(emp_num) FROM professor GROUP BY prof_office` |
| | Count | **Input:** Show the magnitude of unique transaction types.
**Output:** `SELECT COUNT(DISTINCT transaction_type) FROM financial_transactions` |
| Subquery-based Operations | Most-frequent | **Input:** Display the sex and first name of students with the prevalent major.
**Output:** `SELECT Sex , Fname FROM Student WHERE Major IN (SELECT Major FROM Student GROUP BY Major ORDER BY COUNT(*) DESC LIMIT 1)` |
| | Second-maximum | **Input:** Which major has runner-up number of students?
**Output:** `SELECT major FROM (SELECT major , COUNT(*) AS t_prop FROM student GROUP BY major ORDER BY COUNT(*) DESC LIMIT 2 AS t_tab ORDER BY t_tab.t_prop LIMIT 1` |
| | Above-average | **Input:** What are the name of players who got satisfactory points?
**Output:** `SELECT name FROM player WHERE points > (SELECT AVG(points) FROM player)` |
| | Value not present | **Input:** How many customers are absent from having an account?
**Output:** `SELECT COUNT(*) FROM customers WHERE customer_id NOT IN (SELECT customer_id FROM accounts)` |
| | More than max (conditionally) | **Input:** Which songs dominate those with a rating below 6 in terms of resolution?
**Output:** `SELECT f_id FROM song WHERE resolution > (SELECT MAX(resolution) FROM song WHERE rating < 6)` |
| Value-conditioned | 4 credit courses | **Input:** List the names of all heavy courses ordered by their titles and credits.
**Output:** `SELECT title FROM course WHERE credits = 4 ORDER BY title, credits` |
| | Salary more than 30000 | **Input:** Display information on those overpaid employees who joined after 1st April, 1995.
**Output:** `SELECT * FROM employees WHERE salary > 30000 AND hire_date > '1995-04-01'` |
| | Physics and Biology departments | **Input:** What are the addresses of pure-science subject departments? |
**Output:** `SELECT dept_address FROM department WHERE dept_name IN ('Physics', 'Biology')` |
| | Yellow card | **Input:** How many aggressive players have more than 1000 hours of training?
**Output:** `SELECT COUNT(*) FROM player WHERE hs > 1000 AND ycard = 'yes'` |
| | Mountain View and Palo Alto cities | **Input:** How many trips did not end in tech-towns?
**Output:** `SELECT COUNT(*) FROM trip AS t1 JOIN station AS t2 ON t1.end_station_id = t2.id WHERE t2.city NOT IN ('Mountain View', 'Palo Alto')` |
| Column Operations | Concatenation of last and first name | **Input:** How many students are there with 'gE' in alias?
**Output:** `SELECT COUNT(*) FROM student WHERE lname \|\| fname LIKE '%gE%'` |
| | Subtraction of end and start dates | **Input:** What are the unique job ids in job history when tenure is more than 4.
**Output:** `SELECT DISTINCT job_id FROM job_history WHERE end_date - start_date > 4` |
| | Product of Course and Prerequisite IDs | **Input:** How many courses have prerequisite with requirement-id less than 100000?
**Output:** `SELECT COUNT(*) FROM course WHERE course_id IN (SELECT course_id FROM prereq WHERE course_id * prereq_id < 100000)` |

Table 4: Examples of all single-word novel interpretations used in MAGNIFICo. Illustrated examples use a *plausible* English form.

| INTERPRETATION | EXAMPLES |
|---|---|
| Number of characters less than 8 | **Input:** Find the average unit price for a track. Display outputs only if the name is within system length constraints. 
 **Output:** SELECT AVG(unitprice) FROM track WHERE LENGTH(Name) < 8 |
| Value greater than the difference of the average and the standard deviation | **Input:** Find the campuses whose campus fee is in first order outlier range. 
 **Output:** SELECT campus FROM csu_fees WHERE campusfee > (SELECT AVG(campusfee) - STDEV(campusfee) FROM csu_fees) |
| Value less than average | **Input:** What are the name of rooms that have a cost lying in the community-mandated spectrum. 
 **Output:** SELECT roomname FROM rooms WHERE baseprice < ( SELECT AVG(baseprice) FROM rooms) |
| Hire date in July or August 1987 | **Input:** Get the details of employees who manage a department. Show outputs only for those employees that were hired during the months of union labour strike. 
 **Output:** SELECT DISTINCT * FROM employees AS t1 JOIN departments AS t2 ON t1.departmen_id = t2.department_id WHERE hire_date >= '1987-07-01' AND hire_date < '1987-09-01' AND t1.employee_id = t2.manager_id |
| Physicians that are not an intern | **Input:** List the name of board-certified and licensed physicians who never took any appointment. 
 **Output:** SELECT name FROM physician EXCEPT SELECT t2.name FROM appointment AS t1 JOIN physician AS t2 ON t1.physician = t2.employeeid WHERE position NOT IN ('Staff Internist') |
| Number of docks greater than 19 | **Input:** How many biking association compliant stations are in Mountain View? 
 **Output:** SELECT COUNT(*) FROM station WHERE dock_count >= 19 AND city = 'Mountain View' |

Table 5: Examples of all novel interpretations represented by a *phrase* of English words used in MAGNIFICO.

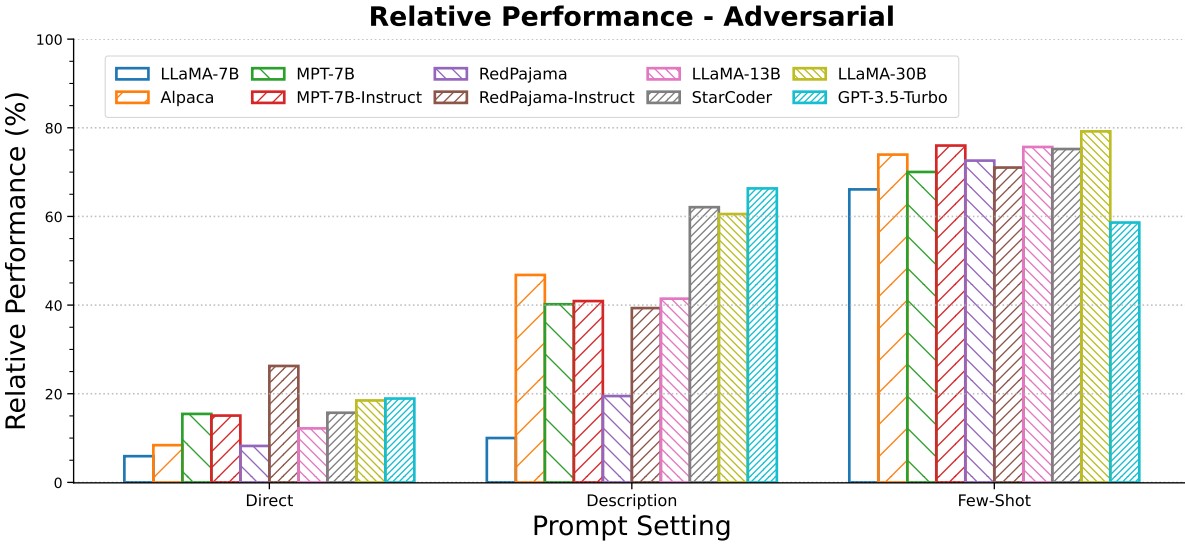

Figure 17: Relative performance (↑) of all models on MAGNIFICO across various prompt settings when the TOKEN is a *adversarial* word.

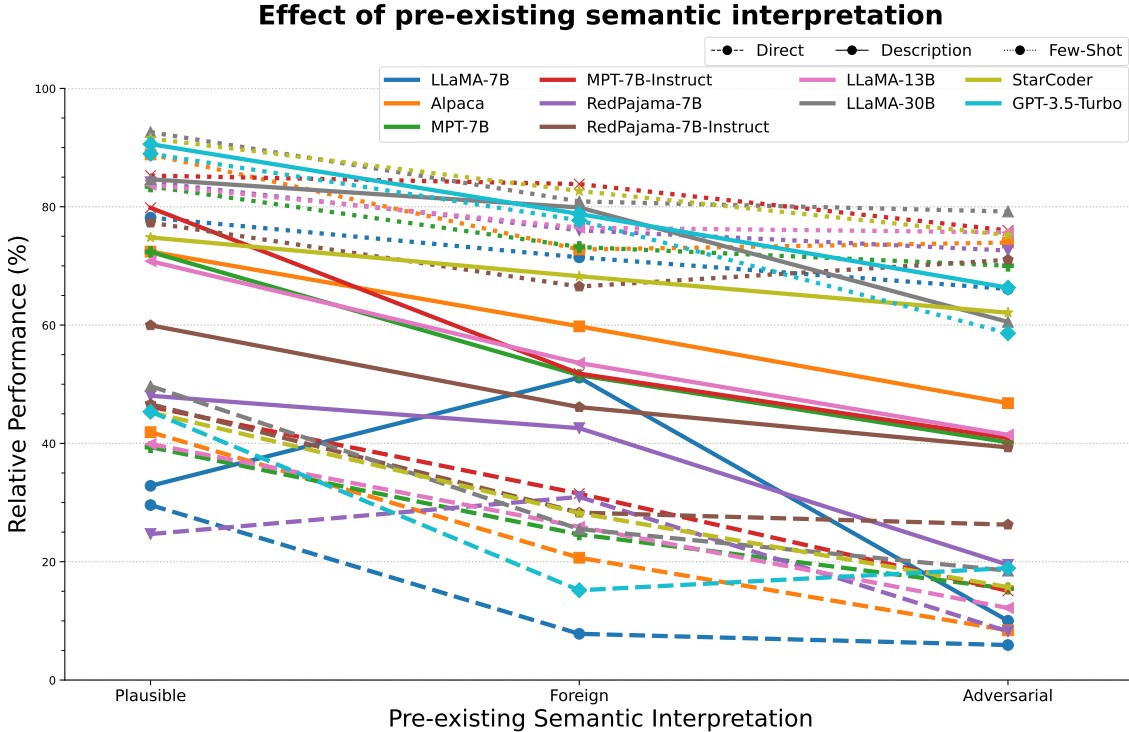

Figure 18: Relative performance (↑) of all models across forms of different types for all prompt settings.

You are DialogueGPT - a tool to generate realistic long-form multi-turn dialogues based on the situation provided.

**User Prompt:**

You are given a database schema. Examples of the data in each of the tables is provided:

```
CREATE TABLE IF NOT EXISTS `departments` (
  `DEPARTMENT_ID` decimal(4,0) NOT NULL DEFAULT '0',
  `DEPARTMENT_NAME` varchar(30) NOT NULL,
  `MANAGER_ID` decimal(6,0) DEFAULT NULL,
  `LOCATION_ID` decimal(4,0) DEFAULT NULL,
  PRIMARY KEY (`DEPARTMENT_ID`)
)
/*
3 example rows:
SELECT * FROM `departments` LIMIT 3;
DEPARTMENT_ID   DEPARTMENT_NAME MANAGER_ID      LOCATION_ID
10              Administration  200             1700
20              Marketing       201             1800
30              Purchasing      114             1700
*/
.
.
.
/*
CREATE TABLE IF NOT EXISTS `jobs` (
  `JOB_ID` varchar(10) NOT NULL DEFAULT '',
  `JOB_TITLE` varchar(35) NOT NULL,
  `MIN_SALARY` decimal(6,0) DEFAULT NULL,
  `MAX_SALARY` decimal(6,0) DEFAULT NULL,
  PRIMARY KEY (`JOB_ID`)
)
/*
3 example rows:
SELECT * FROM `jobs` LIMIT 3;
JOB_ID  JOB_TITLE       MIN_SALARY      MAX_SALARY
AD_PRES President       20000           40000
AD_VP   Administration Vice President   15000   30000
AD_ASST Administration Assistant        3000    6000
*/
```

Generate a 20-turn dialogue between two users of this database. Somewhere near the start of the conversation, user1 says that based on the schema, some people are overpaid. In response to user2 asking what user1 means when they say someone is overpaid, user1 will casually mention that according to them, anyone that earns a salary more than 30,000 is overpaid. The rest of the conversation should make no mention of overpaid. The conversation should not include SQL queries.

Figure 19: Prompt used for generating long form dialogues using GPT-4.

```
CREATE TABLE IF NOT EXISTS `departments` (
  `DEPARTMENT_ID` decimal(4,0) NOT NULL DEFAULT '0',
  `DEPARTMENT_NAME` varchar(30) NOT NULL,
  `MANAGER_ID` decimal(6,0) DEFAULT NULL,
  `LOCATION_ID` decimal(4,0) DEFAULT NULL,
  PRIMARY KEY (`DEPARTMENT_ID`)
)
/*
3 example rows:
SELECT * FROM `departments` LIMIT 3;
DEPARTMENT_ID   DEPARTMENT_NAME MANAGER_ID      LOCATION_ID
10              Administration  200             1700
20              Marketing       201             1800
30              Purchasing      114             1700
*/
.
.
.
/*
CREATE TABLE IF NOT EXISTS `jobs` (
  `JOB_ID` varchar(10) NOT NULL DEFAULT '',
  `JOB_TITLE` varchar(35) NOT NULL,
  `MIN_SALARY` decimal(6,0) DEFAULT NULL,
  `MAX_SALARY` decimal(6,0) DEFAULT NULL,
  PRIMARY KEY (`JOB_ID`)
)
/*
3 example rows:
SELECT * FROM `jobs` LIMIT 3;
JOB_ID  JOB_TITLE       MIN_SALARY      MAX_SALARY
AD_PRES President       20000           40000
AD_VP   Administration Vice President   15000   30000
AD_ASST Administration Assistant        3000    6000
*/

-- Using valid SQLite, answer the following questions for the tables provided above.

-- **The word 'overpay' refers to those with salary more than 30000.**

-- what is all the information about overpaid employees hired before April 2, 1995?
SELECT
```

Figure 20: Example prompt for **Create Table + Select 3** where the prompt contains a description of the novel interpretation.

```
CREATE TABLE IF NOT EXISTS `regions` (
  `REGION_ID` decimal(5,0) NOT NULL,
  `REGION_NAME` varchar(25) DEFAULT NULL,
  PRIMARY KEY (`REGION_ID`)
)
/*
.
.
.

create table prereq
        (course_id              varchar(8),
         prereq_id              varchar(8),
         primary key (course_id, prereq_id),
         foreign key (course_id) references course (course_id)
                on delete cascade,
         foreign key (prereq_id) references course (course_id)
        )
/*
3 example rows:
SELECT * FROM prereq LIMIT 3;
course_id       prereq_id
376     130
760     169
403     345
*/

-- Using valid SQLite, answer the following questions for the tables provided above.

-- on what dates were overpaid employees without the letter m in their first names hired?
select hire_date from employees where salary > 30000 and first_name not like '%m%'

-- give the distinct department ids of departments in which a manager is in charge of one or more
    overpaid  employees?
select distinct department_id from employees where salary > 30000 group by department_id , manager_id
    having count(employee_id) >= 1

-- find the names of all overpaid  instructors in comp. sci. department .
select name from instructor where dept_name = 'Comp. Sci.' and salary > 30000

-- what is all the information about overpaid employees hired before April 2, 1995?
SELECT
```

Figure 21: Example prompt with few-shot examples for the novel interpretation represented by the *plausible* form 'overpaid' followed by the query input.

User1: Hey, I've been working with this database schema and I wanted to discuss some of the tables
    and their relationships.

User2: Sure, I'm familiar with the schema. What do you want to know?

User1: I noticed there's a table for employees with a salary column. Which employees would you
    consider to be **overpaid** ?

User2: Well, that's subjective and depends on various factors. But for the sake of this conversation,
     let's say **anyone earning a salary more than 30,000 is
    overpaid**. What do you want to discuss in relation to the schema?
...
/* conversation containing 2000 words  */
...

User2: The benefits_enrollment table has a foreign key EMPLOYEE_ID, indicating which employee the
    benefit enrollment is associated with. It also has columns for the benefit name, enrollment date,
     and status.

User1: Suppose you are given the following schema:

```
CREATE TABLE IF NOT EXISTS `regions` (
  `REGION_ID` decimal(5,0) NOT NULL,
  `REGION_NAME` varchar(25) DEFAULT NULL,
  PRIMARY KEY (`REGION_ID`)
)
/*
3 example rows:
SELECT * FROM `regions` LIMIT 3;
REGION_ID       REGION_NAME
1       Europe\r
2       Americas\r
3       Asia\r
*/
.
.
.
/*
3 example rows:
SELECT * FROM `locations` LIMIT 3;
LOCATION_ID     STREET_ADDRESS POSTAL_CODE      CITY    STATE_PROVINCE  COUNTRY_ID
1000    1297 Via Cola di Rie    989     Roma            IT
1100    93091 Calle della Testa 10934   Venice          IT
1200    2017 Shinjuku-ku        1689    Tokyo   Tokyo Prefecture        JP
*/
```

Using valid SQLite, answer the following question with the corresponding SQL query:
what is all the information about overpaid employees hired before April 2, 1995?

User2: SELECT

Figure 22: Example prompt which involves a long form dialogue containing the description of the novel interpretation. Note that the truncated section of the dialogue has over 2000 words.