# OpenReview forum: "MAGNIFICo: Evaluating the In-Context Learning Ability of Large Language Models to Generalize to Novel Interpretations"
_EMNLP/2023/Conference — EMNLP 2023 Main_

### Official Review · Reviewer_AoAT · 2023-07-23

**Soundness:** 4

**Excitement:**

4: Strong: This paper deepens the understanding of some phenomenon or lowers the barriers to an existing research direction.

**Paper Topic And Main Contributions:**

This paper investigates the ability of LLMs to adapt to new expressions/meanings through in-context learning. The task considered is mapping natural language text to an SQL semantic representation (i.e., to a query that could be executed on a data base). The paper makes two main contributions: the creation of a dataset to study this problem and an extensive analysis of LLMs using this dataset.

The authors create an evaluation dataset (MAGNIFICo) building on an existing text-to-SQL benchmark (Spider by Yu et al. 2018). They define a set of 24 meanings and map them to three types of textual form (word of phrase): an existing word for which the new meaning is plausible, a nonce word, and an existing word with contrary meaning (adversarial).

This dataset is then used to evaluate a range of LLMs in an in-context learning setup that considers three types of prompts: Direct (zero-shot introduction of the new interpretation within the query in natural language), Description (the prompt also includes a natural language definition of the new interpretation), and Few-shot (the prompt includes 5 examples of language-SQL pairs illustrating the new interpretation, without any explicit definition). In addition, for a limited number of instances, the prompt includes a description of the new interpretation within the context of dialogue between two simulated users.

The results show that most LLMs under scrutiny can recover novel interpretations well in the Description and Few-shot setups when the forms are plausible, and that there is a decrease in generalisation ability particularly when the forms are adversarial. When the description is included as part of a dialogue, the performance is lower (with plausible forms) and exhibits recency bias (the results improve when the description is introduced near the end of the dialogue/context window). Finally, there results show that models struggle when multiple interpretations are introduced within a single query and when the forms consists on multi-word phrases rather than single words.

**Questions For The Authors:**

Question A: It was not clear to me how the Descriptions have been generated. Is this actually explained in the paper and I missed it? If it's not, can you please explain?

Question B: In the case where the expression (form) is a phrase rather than a single word, is the distinction between plausible, nonce, and adversarial also made?

Question C: What was the cost of using models that are not freely available to make this work possible (e.g., using GPT-4 to generate the dialogues or analysing the output of a non-free models)? (I find this relevant for assessing the limitations and ethical consequences of the work)


**Reasons To Accept:**

Strengths:

- Novel way to re-frame the problem of word learning and compositional generalisation in LLMs
- Creation of an evaluation dataset with clear and interesting design choices
- Solid experimental design
- Extensive analysis and comparison of a substantial amount of LLMs that differ with respect to model size and the use or not of instruction training
- Clearly written


**Reasons To Reject:**

The paper naturally has some weaknesses, but no major reasons for rejection. See the box on "Presentation Improvements" below for a few things that should be improved.


**Reproducibility:**

3: Could reproduce the results with some difficulty. The settings of parameters are underspecified or subjectively determined; the training/evaluation data are not widely available.

**Reviewer Confidence:**

4: Quite sure. I tried to check the important points carefully. It's unlikely, though conceivable, that I missed something that should affect my ratings.

**Typos Grammar Style And Presentation Improvements:**

- The related work section is too concise. In particular, I miss some more background on text-to-SQL. (I realised later that there is an appendix with further related work, which is not mentioned in the main paper. Space constraints are not an excuse here: the paper does not use space very efficiently.)

- The Spider dataset is not described and this would help to understand better the starting point for the proposed dataset.

- At the beginning of the Results section, I would find it natural to comment on the results obtained in the Direct zero-shot setting.

- The font of pretty much all figures is too small. I wasn't able to read many of the legends and text on a print out of the paper.

---

> ### Author Rebuttal · Authors · 2023-08-29
>
> We thank you for your time spent in reviewing our paper and providing helpful comments. Please see our response below to your questions:
>
> Answer A: The descriptions for each interpretation were manually written by one of the authors and verified by another.
>
> Answer B: No, this distinction is only made for single-word forms.
>
> Answer C:
> We used GPT-4 for:
> (1) constrained paraphrasing: we estimate usage of ~50,000 tokens, which would amount to a cost of not more than \\$3.
> (2) generating dialogues: we estimate usage of ~600,000 tokens for creating the 125 dialogues, which would amount to a cost of not more than \\$30.
> We used GPT-3.5-Turbo for evaluation:
> We estimate ~2M output tokens, which would cost ~\\$4 and ~30M input tokens, which would cost ~\\$45.
>
> In total, we estimate the OpenAI API costs to be ~\\$80-\\$100.

---

### Official Review · Reviewer_9Hzn · 2023-08-02

**Typos Grammar Style And Presentation Improvements:** Finetuning and Fine-tuning should hav…
**Soundness:** 4

**Excitement:**

4: Strong: This paper deepens the understanding of some phenomenon or lowers the barriers to an existing research direction.

**Paper Topic And Main Contributions:**

This work investigates the ability of LLMs to learn and comprehend novel interpretations using in-context learning within the limitations of their knowledge cutoff and the high cost of repeated fine-tuning.

The authors introduce a new evaluation tool called MAGNIFICO, developed within a text-to-SQL semantic parsing framework. This tool is designed to incorporate diverse tokens and prompt settings to imitate the complexity of real-world scenarios.

The authors conducted experimental studies on MAGNIFICO to assess the LLMs' capacity for understanding new interpretations from natural language descriptions, long conversations, and compositional generalization. The findings reveal that LLMs still need to improve in understanding unfamiliar words or combining multiple novel interpretations simultaneously in the same example.

Moreover, the authors analyze the semantic predispositions in LLMs and explore the impact of recency bias when information is presented in lengthy contexts. This suggests that recent details may have a more significant influence on the models' understanding and interpretation.

The authors conclude that there is a need for further development and improvement in these areas, especially in handling the complexity and novelty of interpretations, suggesting that their work can serve as a basis for future research on improving LLMs' in-context learning capabilities.

**Questions For The Authors:**

1. Why didn't you evaluate the GPT-4 model with the MAGNIFICO tool?

**Reasons To Accept:**

1. They present a novel evaluation suit called MAGNIFICO which is applied to assess the in-context learning capabilities of LLMs. This tool could be valuable to other researchers in the field.

2. They extensively investigate how LLMs interpret and understand new words and phrases in different context settings, providing useful insights about their limitations and capacities.

3. The paper identifies key areas where LLMs struggle, such as understanding unfamiliar words or multiple novel words simultaneously. Identifying these areas could guide future improvements in the development of LLMs.

4. This study evaluates the performance of different LLMs which leads to a comprehensive understanding of LLM capabilities.

**Reasons To Reject:**

1. Most importantly, as mentioned in the "limitation" section, the number of test data used for some experiments is inadequate to conclude or make a strong analysis, which weakens the analysis for those subsections. Since the data generation is automated, extending the dataset is necessary.
2. While clearly having access to  GPT-4, this work should have compared their results with the recent GPT-4 model. Their work would have been much more substantial with the latest models in their results.
3. Their tool is built to assess the LLMs while only covering conversational large language models.
4. The conclusion in line 542 regarding 'instruction fine-tuning' doesn't have clear supporting evidence and needs more clarification.

**Reproducibility:**

4: Could mostly reproduce the results, but there may be some variation because of sample variance or minor variations in their interpretation of the protocol or method.

**Reviewer Confidence:**

3: Pretty sure, but there's a chance I missed something. Although I have a good feel for this area in general, I did not carefully check the paper's details, e.g., the math, experimental design, or novelty.

---

> ### Author Rebuttal · Authors · 2023-08-29
>
> We thank you for your time spent in reviewing our paper and providing helpful comments. Please see our response below to specific comments:
>
> *this work should have compared their results with the recent GPT-4 model*
>
> At the time of submission, we were unable to run **evaluation** experiments with GPT-4 because of rate and resource limitations (experimenting on multiple prompt and form settings becomes very expensive). We prioritized the use of GPT-4 for the more important task of helping **create the MAGNIFICo benchmark**.
>
> *Their tool is built to assess the LLMs while only covering conversational large language models*
>
> GPT-3.5-turbo is the only conversational LLM covered in our analysis. All other models that we experimented with (StarCoder, LLaMA, RedPajama, MPT, and RWKV) are general LLMs.

---

### Official Review · Reviewer_jQoo · 2023-08-09

**Typos Grammar Style And Presentation Improvements:** 1. Very minor point for Figure 1, the…
**Soundness:** 5

**Excitement:**

4: Strong: This paper deepens the understanding of some phenomenon or lowers the barriers to an existing research direction.

**Paper Topic And Main Contributions:**

This work collects a text-to-SQL dataset containing words/phrases that have a new meaning(interpretation), and uses it to conduct a very comprehensive study of how well LLMs'  generalize to the novel interpretation under the in-context learning setting.

The author come up with 24 interpretations, e.g. the word "prevalent" can have an interpretation as "Most-frequent".  Based on those interpretations, the author retrieve samples from Spider containing content that can apply the new interpretation with three different methods, and also manually verify the dataset obtained.

The author then tests various LLMs on the task in zero-shot, few-shot, and a setup including additional text description. A list of research questions is proposed and answered based on those results.

Main Contribution

1. Created a dataset that can benchmark LLM's ability on generalizing to new definitions of words. As the generated SQL results can be evaluated in terms of execution accuracy, it has a more clear definition of performance than free-text generation. Manual check also ensures its quality.

2. Well-conducted analysis on how different in-context learning settings affect the model's ability to generalize to novel interpretation.
Further explored other experiment settings, such as multiple new interpretations in one sentence, new interpretations through dialogue, and multiple new interpretations in one sample.

**Questions For The Authors:**

A: Line 336, how does the selection of few-shot examples affect the results? How large is the variance based on your experience?

**Reasons To Accept:**

1.  This work collected a new benchmark dataset that can test for LLMs' in-context learning ability on new meanings of words. The task is useful as it can be an indicator of LLMs' in-context learning ability in general, and it has execution accuracy as a clear metric for performance. The method to collect the data is well articulated so it can scale. A bonus point is that the author checked all the samples collected.
2. The analysis is very through and answered a lot of questions regarding in-context learning, which includes introducing new info with text description, few-shot examples, and dialogue. The paper also discusses the impact of description and LLMs' semantic info obtained during the pretraining, etc.
3. The paper is well written, with plentiful examples in the appendix showing what is done in each stage. The illustrations are clear and visually pleasing. The experiment section is structured in a way to answer different research questions, which makes the results very clear.

**Reasons To Reject:**

1. Although mentioned in section 3, it is still not very clear why SQL is chosen as the task, as the justification is quite general. Despite being easy to evaluate, SQL task restricts the range of new interpretations as it has to be a valid condition for a query.

**Reproducibility:**

5: Could easily reproduce the results.

**Reviewer Confidence:**

4: Quite sure. I tried to check the important points carefully. It's unlikely, though conceivable, that I missed something that should affect my ratings.

---

> ### Author Rebuttal · Authors · 2023-08-29
>
> We thank you for your time spent in reviewing our paper and providing helpful comments. Please see our response below to specific comments:
>
> *it is still not very clear why SQL is chosen as the task*
>
> Our main reasons to work with the text-to-SQL semantic parsing task were:
> (1) It is a fundamental NLU task that has been widely studied.
> (2) It illustrates real-world scenarios in which an automated system must learn novel interpretations.
> (3) Many existing LLMs are able to achieve good performance on the task via in-context learning.
>
> *how does the selection of few-shot examples affect the results? How large is the variance based on your experience?*
>
> Our initial analysis did not reveal much variance in the results when using different sets of few-shot exemplars. We did not run a comprehensive quantitative analysis due to time and resource constraints during submission, but plan to include that in the final version.

---

### Official Review · Reviewer_wHLA · 2023-08-11

**Typos Grammar Style And Presentation Improvements:** NA
**Soundness:** 4

**Excitement:**

3: Ambivalent: It has merits (e.g., it reports state-of-the-art results, the idea is nice), but there are key weaknesses (e.g., it describes incremental work), and it can significantly benefit from another round of revision. However, I won't object to accepting it if my co-reviewers champion it.

**Missing References:**

NA

**Paper Topic And Main Contributions:**

The authors studied the ability of LLMs to acquire novel interpretations using in-context learning. They considered three different scenarios (In the context of text-to-sql semantic parsing). In the first scenario, they considered providing a description of the novel term. In the second scenario, they illustrated the term via few shot samples. In the third scenario, they considered understanding the interpretation from long-from dialogue. The contributions of the paper are:
- magnifico benchmark: a benchmark in which each sample is a text-to-sql semantic parsing problem that require models to understand one or more novel interpretations
- Analysis on 11 different LLMs in which they showed that models are highly capable in learning novel interpretations.
-  Analysis on the limitations of LLMS where they showed that such models fail at learning multiple novel interpretations simultaneously however, they fail at learning multiple novel interpretations at once.

**Questions For The Authors:**

Question A: How do you define what a "novel" interpretation is?

Question B: How did you choose the interpretations that you want to include in the the benchmark?

Question C: How many annotators verified the ability of gpt-4 in generating conversations?

Question D: what is the motivation behind using foreign forms?

**Reasons To Accept:**

- Providing a benchmark to test the ability of LLMs in learning novel interpretations.
- Interesting analysis about the capabilities and limitations of LLMs in multiple settings

**Reasons To Reject:**

- Restricted the analysis to one task only which is not clearly motivated.
- Results are scattered between the body and appendix and the reader must refer to the appendix quite often.

**Reproducibility:**

3: Could reproduce the results with some difficulty. The settings of parameters are underspecified or subjectively determined; the training/evaluation data are not widely available.

**Reviewer Confidence:**

3: Pretty sure, but there's a chance I missed something. Although I have a good feel for this area in general, I did not carefully check the paper's details, e.g., the math, experimental design, or novelty.

---

> ### Author Rebuttal · Authors · 2023-08-29
>
> We thank you for your time spent in reviewing our paper and providing helpful comments. We are glad that you recognise the utility of our benchmark and find our analysis about the capabilities and limitations of LLMs interesting. Please see our response below to specific comments:
>
> *Restricted the analysis to one task only which is not clearly motivated*
>
> We chose the text-to-SQL semantic parsing task because:
> (1) It is a fundamental NLU task that has been widely studied.
> (2) It illustrates real-world scenarios in which an automated system must learn novel interpretations.
> (3) Many existing LLMs are able to achieve good performance on the task via in-context learning.
>
> *Results are scattered between the body and appendix and the reader must refer to the appendix quite often*
>
> We believe that the current body of the paper cohesively presents all main results and discussions along with necessary background and details. We used the appendix to provide auxiliary information regarding the exact prompt formats and dataset creation. Results of some models such as RWKV and RedPajama as well as some additional related works were shifted to the appendix because of lack of space - we shall add these to the main body in the final version.
>
> Ans A. A “novel” interpretation in the context of our work refers to a part of the language input that has to be interpreted in a specific well-defined manner (often in a way that the LM would not have seen during pretraining) when generating the output SQL query.
>
> Ans B. We defined broad categories of interpretations based on patterns that were repeated in the SQL queries in existing text-to-SQL datasets (See Table 1). We then defined each interpretation as a specific instantiation of one of these categories, influenced by the quantity of data that could be generated for it.
>
> Ans C. All GPT-4 generated conversations were manually reviewed by one author.
>
> Ans D. Our experiments with foreign form illustrate a scenario where a system needs to learn a word that is not present in its vocabulary (for e.g., a new word coined after the system was trained) and therefore has no existing semantic disposition for that token.
>
> In light of our response above and other reviews, could you kindly reconsider your evaluation?

---

### Meta-Review · Area_Chair_Xceq · 2023-09-16

**Recommendation:** 5

**Metareview:**

This paper focuses on the ability of LLMs to learn nover interpretations of word and phrases via in-context learning. Specifically, they introduce a new text-to-SQL semantic parsing  dataset called MAGNIFICo which contains diverse tokens and prompt settings. Theirexperiments indicate that LLMs do resonably well on this task but there is still room for improvement.

Overall, reviewers agree that the task is interesting and the paper is well-written with analysis and experiments. While it does have some drawbacks ((1) only one task for studying a broader phenomenon, and (2) fewer data points for some settings), the contributions are above the threshold for acceptance.

---

### Decision · Program_Chairs · 2023-10-07

**Decision:**

Accept-Main

**Comment:**

This paper focuses on the ability of LLMs to learn nover interpretations of word and phrases via in-context learning. Specifically, they introduce a new text-to-SQL semantic parsing  dataset called MAGNIFICo which contains diverse tokens and prompt settings. Theirexperiments indicate that LLMs do resonably well on this task but there is still room for improvement.

Overall, reviewers agree that the task is interesting and the paper is well-written with analysis and experiments. While it does have some drawbacks ((1) only one task for studying a broader phenomenon, and (2) fewer data points for some settings), the contributions are above the threshold for acceptance.